# Wetland Fire Scar Monitoring and Its Response to Changes of the Pantanal Wetland

**DOI:** 10.3390/s20154268

**Published:** 2020-07-31

**Authors:** Xiaodong Li, Kaishan Song, Ge Liu

**Affiliations:** 1Shandong Key Laboratory of Eco-Environmental Science for Yellow River Delta, Binzhou University, Binzhou 256603, China; lixiaodong2020@bzu.edu.cn; 2Key Laboratory of Wetland Ecology and Environment, Northeast Institute of Geography and Agroecology, CAS, Changchun 130102, China; songks@iga.ac.cn

**Keywords:** wetland fire scar, Pantanal wetland, ecological response, Sentinel-2A, dynamic ratio

## Abstract

Fire is an important disturbance factor which results in the irreversible change of land surface ecosystems and leads to a new ecological status after the fire is extinguished. Spanning the period from August to September 2019, the Amazon Forest fires were an unprecedented event in terms of the scale and duration of burning, with a duration of 42 days in the Pantanal wetland. Based on the observation data of wildfire and two Sentinel-2A images separated by a 35-day interval, the objectives of this study are to use the Normalized Burn Ratio (NBR) to map the spatiotemporal change features of fire and then quantitatively measure the fire severity and the impact of fire on the Pantanal wetland. The overall accuracy and Kappa coefficient of the extracted results of wetland types reached 80.6% and 0.767, respectively, and the statistically analyzed results showed that wildfires did not radically change the wetland types of the Pantanal wetland, because the hydrological variation of the burned area was still the main change factor, with a dynamic ratio of ≤50%. Furthermore, the savanna wetland in the burned area was the wetland type which was most affected by the fire. Meanwhile, fire scars belonged to the moderate and low-severity burned areas, with a maximum burn area of 599 km^2^. The case enriches the research into the impact of wildfire as the main disturbance factor on the change of wetland types and provides a scientific reference for the restoration and sustainable development of global wetland ecosystems.

## 1. Introduction

Commonly, “wetland fire” refers to naturally occurring wildfires, which are considered one of the most important environmental disturbing factors [1,2,3,4]. In general, wildfires have negative environmental effects such as global warming and damage to land fertility, ecological diversity, land use and property, but they also have positive impacts on vegetation regeneration and nutrient recycling [5,6,7,8]. With the increasingly frequent wildfires in South America during the last 15 years, grasslands and savannas present a high mortality rate. Meanwhile, the spread of fire has greatly accelerated, leading to more open land-cover types through continuous grass layers characterized by grasslands and savannas [9]. Fire has therefore been recognized as a common tool to remove biomass from newly cleared fields [10] and as a management tool [11].

Fire disturbance is closely related to the balance of ecosystems. According to the research into of the causes of fire, natural and human factors that lead to the occurrence of fire include long drought [12], rapid land-use change [13] and deforestation [14]. In particular, the slash-and-burn agriculture caused by human activities has been a new disturbance to the land-cover types in Amazon, with increasing sources of ignition and a more flammable landscape structure [15]. If deforestation continues unabated, 16% of the region’s remaining forests will likely burn by 2050 [16].

The effects and functions of fire are different in different ecosystems [17,18]. The study into the post-fire impact on ecological factors includes the following aspects: (1) wildfire increases the probability of flash floods and the debris flows of the wetland [19,20]. However, the change of the hydrological response after fire does not always decrease gradually and linearly with time; a number of post-fire studies have noted that low-intensity rainfall events can result in a minimal initial post-wildfire runoff response [21,22,23]. (2) The reshaping of the ecological structure is a direct change feature in the post-fire period. Combined with post-fire climate change, wildfire—as a main disturbance—will determine the future land-cover types and whether these cover-types can sustain critical habitat types and ecosystem services over the long term or will enter a common period of degradation [24]. (3) The ecologically fragile zone is the direct result of fire on the land’s ecological environment. Experimental data collected from the southeast Amazonia showed that post-fire annualized tree mortality rates jumped from 10% to 90% when wildfires occurred along the forest–grass transition zone during drought years compared with non-drought years [25]. In other words, when severe droughts occur in the ecological landscape transition zone, fires can be catastrophic and have long-lasting ecological effects [26]. (4) Wildfires have also been reported to have reduced the number of birds [27] and dung beetles [28] in the Amazonia forest. On the other hand, animal groups which survive in degraded forests can help to promote forest recovery [29,30].

With the emergence and development of remote sensing technology, the vast amount of available satellite image data has become an important means of obtaining fire information and monitoring the change of wetland types in fire areas, saving both time and effort [31,32]. In particular, with the improvement of the spatiotemporal resolution and spectral resolution of remote sensing data, remote sensing technology can better realize the spatiotemporal continuous monitoring of the surface change process, providing a possible method for the study of wetland change detection after fire [33,34,35,36,37,38].

The main objectives of this study are to (1) extract the burned area of the Pantanal wetland, and (2) analyze the spatiotemporal change features of the wildfire and its impact on the wetland types in the Pantanal. Specifically, this study aims to quantitatively measure the fire’s severity and the response of wetlands to the fire by mapping wetland types as well as their changes based on open-access remote sensing data resources. The analysis of the relationship between fire and the change of wetland types can enrich the research into the influence of wildfire as the main disturbance factor for changing wetland types and provide a scientific reference for the restoration and sustainable development of wetland ecosystems.

## 2. Study Area and Source Data

### 2.1. The Pantanal Wetland

The Pantanal wetland, as the largest tropical inland plain wetland in the world, is mainly located in the South Mato Grosso and stretches to the southern part of Mato Grosso in Brazil; the wetland is also partly in Bolivia and Paraguay [39,40]. The Pantanal wetland has the highest diversity of aquatic plants and is one of the largest freshwater wetland ecosystem in the world [41]. The study area is a northern sub-region of the Pantanal, called Corixo Grande (CORI), with an area of 2381.4 km^2^. As a part of the Pantanal wetland, the geographic longitude range of the study area is located at 58°30′–57°18′ W, and the latitude range is from 17°48′ to 16°30′ S (Figure 1). The period from December to May of the following year is the rainy season of Pantanal with a rainfall of 1000 to 1500 mm. In the rainy season, 80% of Pantanal will be flooded, forming a unique seasonal floodplain ecosystem that provides an ideal habitat for a large number of endangered species and birds [42].

### 2.2. Wetland Fires

The Visible Infrared Imaging Radiometer Suite (VIIRS) instrument on-board of the joint NASA/NOAA Suomi National Polar-Orbiting Partnership (Suomi-NPP) satellite provides the day/night information of wildfires with fine spatial resolution imagery (375 m). Meanwhile, the Moderate Resolution Imaging Spectroradiometer (MODIS) provides the daytime visible imagery and infrared night-time imagery of global fires with spatial resolution imagery (1000 m). In these images, the fire activity or hot spots (i.e., fire scars) detected by the thermal bands are outlined as red bands. A fire scar is an area that is easily detected by the thermal detectors on the satellite instrument due to the higher land temperature. Such a burned area is diagnostic for the detection of fire, even if not accompanied by plumes of smoke [43]. The VIIRS and MODIS tools can also provide daily vector data for global fire observations (Table 1).

### 2.3. Satellite Data

On 23 June 2015, the European Space Agency (ESA) successfully launched Sentinel-2, which is used for environmental monitoring. Carrying an optical sensor, the Sentinel-2, as the second satellite of Europe’s Copernicus plan, can obtain imagery with a high spatial and temporal resolution through its combination with Sentinel-1A. The functional improvement of Sentinel-2 broadens the monitoring scope and improves the environmental monitoring ability of Sentinel-1A and can be utilized for food safety, forest monitoring, land use change and vegetation health monitoring, etc.

Sentinel-2 Multispectral Instrument (MSI) data contain 13 spectral bands, with a spatial resolution of 10 m and a revisit period of 10 days [44]. From visible and near-infrared to short-wave infrared bands, Sentinel-2A imagery is unique in that it contains three bands in the red-edge range, which is very effective for monitoring vegetation health information. Sentinel-2A multispectral data are selected to monitor the actual process and specific situation of fires occurring in the study area (Table 2).

Remote sensing imagery processing is necessary to obtain the calculated data for the further extraction of the burned area information in the study area, including spatial registration, geometric correction, atmospheric correctio, and image clipping. Finally, all map parameters of the datasets were transformed into the World Geodetic System 1984 (WGS-84) geographic coordinate system and the Universal Transverse Mercator (UTM) projection [45].

## 3. Methods

To investigate whether the Sentinel-2A data can capture the short-term burning events and seasonal dynamics of wetlands, the analysis processes included the calculation of the Normalized Burn Ratio (NBR) and the dynamic ratio, as described below for the Sentinel-2A datasets (Figure 2).

### 3.1. Mapping the Distribution of Fire Scars

A wildfire which leads to a change of wetland type is not only an important disturbance factor for ecosystems but also a significant influence factor of global atmosphere quality. In order to deliver rapid information about areas damaged by fires, the NBR and the relative extensible algorithms have been applied for the regional cartography of burned areas based on high-resolution optical satellite data [46,47]. The formula for the calculation of NBR is shown in Equation (1).
NBR = (NIR − SWIR2)/(NIR + SWIR2)(1)

Similar to the formula of the Normalized Difference Vegetation Index (NDVI), the NBR uses the near infrared (NIR) and short-wave infrared 2 (SWIR2) wavelengths, which correspond to band 8 (0.76–0.90 μm) and band 12 (2.07–2.32 μm), respectively.

The fire severity levels include low-severity burn, moderate-severity burn and high-severity burn, which were classified based on the USGS (United States Geological Survey) Fire Effects Monitoring and Inventory Protocol (FireMON) program. The range derived from the difference of NBRs (dNBR) indicates a burn severity from 0.1–0.27, 0.27–0.66 and greater than 0.66, respectively [46].
dNBR = NBR _pre_ − NBR _post_(2)
where NBR _pre_ and NBR _post_ correspond to the landscape ecological status of the pre-fire and the post-fire, respectively. The burn severity index is a good estimation for the measurement of the ecological change in land conditions post-fire; thus, the dNBR was calculated by subtracting the post-fire NBR from the pre-fire NBR.

### 3.2. The Change Patterns of Wetland in the Burned Area

#### 3.2.1. Selection of Wetland Characteristic Indicators

The Modified Normal Differential Water Index (MNDWI) was used to extract water information from the land surface. Combined with other indexes, the Water Body Index can be used to construct multidimensional feature data sets, which can quantitatively measure the ecological changes of wetlands [48].
MNDWI = (Green − SWIR1)/(Green + SWIR1)(3)

The Normalized Difference Soil Index (NDSI) is also known as the Conventional Soil Index. The ratio is based on the near infrared (NIR) and short-wave infrared 1 (SWIR1, 1.57–1.65 μm) bands of remote sensing image data. The negative value of NDSI represents water body areas [49].
NDSI = (SWIR1 − NIR)/(SWIR1 + NIR)(4)

The Normalized Difference Vegetation Index (NDVI) is the ratio of the reflectance between the near-infrared and red bands [50]. NDVI is calculated by the pixel-based method using Equation (5).
NDVI = (NIR − R)/(NIR + R)(5)

NDVI is mainly used to detect vegetation growth status. NDVI has a range from −1 to 1, and positive values indicate that the vegetation increases with the improvement of vegetation coverage; negative values indicate that the surface cover type is cloud, water, snow, and rock, etc. [51].

In Equations (1)–(5), Green represents the reflection of the green band (0.52–0.60 μm), R represents the reflection of the red band (0.63–0.70 μm), NIR is the near infrared band (0.76–0.90 μm) and SWIR1 is the short-wave infrared band (1.57–1.65 μm).

#### 3.2.2. Dynamic Ratio of Wetlands in the Burned Area

The dynamic ratio method, as a land cover change detection method, is used to monitor the change features of wetland ecological factors following wetland fires, based on Sentinal-2A time series data. The mean value of the ecological indexes of two images is used to represent a constant status at the same location, and the difference of images is used to represent the variation range of the ecological status. The dynamic ratio based on the standard deviation and the mean is denoted as the change of ecological factors in the Pantanal wetland [52].
Dynamic ratio = (x_2_ − x_1_)/Mean(x_1_, x_2_)(6)
where x_1_ and x_2_ correspond to the landscape ecological status of two multispectral images. If only the changes of the data before and after the two periods are required to be observed—that is, when the sample size is 2—the rate of dynamic change is actually a quantitative measure of the increase or decrease of the sample in the two periods. 

#### 3.2.3. Wetland Types and Accuracy Evaluation

Data of land cover are referenced from the global surface coverage remote sensing data products (GlobeLand30). The main wetland types in the remote sensing classification in the study area include constant water bodies, marsh (swampy grassland), and savanna wetland (OWSF (open wood savanna–flood)). The Sentinel-2A multispectral images in January, July, August and September 2019 with a spatial resolution of 10 m were utilized as the main data source for the classification and combined with the auxiliary data for the accuracy evaluation, which mainly included high-resolution Google Earth images. Supervised classification was used to classify the images based on the training sets provided by the mapper’s professional knowledge and the field survey. The classification algorithm used by this research is the support vector machine (SVM), because the SVM algorithm can achieve better classification accuracy based on the small samples selected from the study area [53]. The training data given by the user guides the software regarding the types of pixels to be selected for certain land cover types.

To evaluate the applicability of the Sentinel-2A images, 1784 points in the core areas were randomly collected and extracted by compute, and then superimposed with a high-resolution satellite image from Google Earth^®^ for the accuracy test. Furthermore, the selected samples were randomly divided into two parts: 70% of the samples were used for the classification and 30% of the samples were used for the accuracy evaluation. The main quantitative evaluation of the results included the overall accuracy (OA) and Kappa coefficient (KC) [40]. Finally, 535 sample points were used for the accuracy evaluation of the classification results, accounting for 30% of the total sample size.

## 4. Results

### 4.1. Wetland Fire Sites and Wetland Fire Dynamic

(1)The Scope of Fire Sites

Wetland fire scars, known as burned or burning fields, are the form of the spatial expansion of fire points. Based on the premise of ensuring the continuity and integrity of the spatial distribution of fires, the Pantanal wetland’s burned and burning fields in July, August and September were manually interpreted and extracted by a mapper combined with a computer. Then, a total of 30 burned or burning fields was divided from the overfired area of the Pantanal wetland park, among which 7, 12, and 11 fire fields were found in July, August and September, respectively. Finally, the extracted results of each month were superimposed to obtain a comprehensive map from July to September 2019. 

According to the center-location distribution data of fire scars (Figure 3) from July to September 2019, the direct impact of Amazon forest fires on wetlands was mainly distributed in the central part of the Pantanal wetland and the east of Rio Corixa Grande. The scale of combustion showed the largest range in September, accounting for 26.3% of the total study area. 

(2)Fire Severity

Firstly, the combustion indexes (NBR) for January, July, August and September were calculated based on the Sentinel-2A images, respectively. Then, the monthly differences of combustion indexes (dNBR) were calculated and cut using the fire region vector data of the corresponding month to obtain the monthly results in the burned area. Finally, the monthly cut results were superimposed to obtain a comprehensive map (Figure 4) of the main months from July to September 2019.

The stacking layer of wetland fire severity is obtained using the NBR calculated from Sentinel-2A images with the spatial resolution of 10 m and the spatial distribution map of wetland fire sites provided by MODIS and VIIRS from July to September 2019.

Conventional indexes, including the NBR and dNBR, were used to quantitatively measure the fire severity of the study area. In July, the burned area was relatively small, with an area of 13.2 km^2^. In August, the difference of the combustion indexes (dNBR) and the burned area showed the most obvious change, at 0.203 ± 0.122 and 264.7 km^2^, respectively. In September, the burned area reached its maximum range of 321.1 km^2^ during the observation period. Meanwhile, the NBR was the lowest (−0.162 ± 0.146) (Table 3).

The difference calculation results provided by this paper are the superimposed map of the fire-region in August and September. In August, the low and moderate-severity burned areas accounted for 67.06% and 32.75% of the combustion area, respectively (Figure 4). The burnt areas caused by fire continued to expand during the observation period in September; 53.42% of the combustion area was low-severity burnt area, while 46.48% was moderate-severity burnt area.

### 4.2. Correlation between the Fire and Wetland Type

#### 4.2.1. Maximum Burn Range of Fire and the Wetland Type

The maximum burn range of fire in the monitoring area was determined by the sum of all fire-scars that occurred in the duration of monitoring from July to September. Furthermore, the main analyzed content was the change of wetland types in the maximum burn range disturbed by the fire.

The wetland types in January 2019 were taken as the reference data for unburned area and used for the comparison analysis of the change of land-cover types, especially the wetland types including marsh, savanna wetland and water. The finally results showed that savanna wetland was the main impacted type in the occurrence process of wildfire, accounting for 24.8% of the maximum burn range of fire. With the significant expansion of fire fields in September 2019, the increased bared land accounted for 90.8% of the maximum burn range (Figure 5).

According to the statistical analysis of the average percentage of wetland types in the maximum burn range of the fire from July to September 2019, the main land cover-type in the maximum burn range was Savanna wetland, which was the main wetland type impacted by the fire, with an average percentage of 20.21 ± 9.54% (Figure 6). Meanwhile, forest and bared land were the main non-wetland types affected by the fire, with average percentages of 3.2 ± 2.18% and 72 ± 12.8%, respectively.

#### 4.2.2. Fire Severity and Wetland Type

(1)The percentage of wetland/non-wetland types with different fire severities

Due to the burning area expanding rapidly in September, the change of wetlands in the study area was mainly manifested in terms of two aspects: the percentage of wetland and the wetland type change.

The zonal statistics of wetland types in the burned area were determined based on the ArcGIS platform. According to the final analyzed results, the burned marsh area accounted for 0.23% of the total fire fields in August, and the burnt savanna wetland area accounted for 33.8%. With the expansion of burned area in September, the overfired area of savanna wetland, as the main wetland type affected by fire, was 119 km^2^ (24.9%), 14.7% of which belonged to the moderate-severity burnt area (Figure 7). 

First of all, the percentage of savanna wetland in the burned area continued to decrease after entering the dry season, as did the actual water area affected. However, the overfired area of marsh expanded to 1.74 km^2^, accounting for 0.36% of the total burned area. In conclusion, savanna wetland was shown to be the most significant response factor in the process of the fire disturbance. Meanwhile, the increase of bared land as the final result of the fire in the study area, accounting for percentages of 63% of the burned area in August and 71% of the burned area in September (Figure 7).

(2)The transfer-in/-out wetland types in different fire-severity areas

Considering the small area of fire fields in July, we focused on and analyzed the main wetland types in the land-cover transformation process of the study area in August and September.

The results showed that the transfer-out percentages of savanna wetland in August 2019, under different fire-severity areas, were 63.14% in the area of fire-severity Level-1 and 73.14% for Level-2/3), respectively. Similarly, the change percentages of savanna wetland in September were 72.13% for Level-1 and 88.45% for Level-2/3, respectively. Meanwhile, the main transfer-out type in the fire process in September 2019 was the savanna wetland, with a transfer-in ratio of 2.2% for Level-1 and 0.39% for Level-2/3 (Table 4). The main transfer-in type in September 2019 was the bared land, with a high transfer-in ratio of 89.43% for Level-1 and 97.97% for Level-2/3 (Table 5). Considering the total of change pixels, the main burning process of the Pantanal wetland occurred in September 2019. In addition, savanna wetland was the main affected wetland type of moderate and low-severity burn areas in August, with conversion percentages of 36.39% for low-severity and 51.05% for moderate-severity areas. In September, with the increased number of fire sites and the expanded fire area, savanna wetland was still the main influenced wetland-type of the moderate and low-severity burned areas. In conclusion, fire fields in the monitoring area belonged to the moderate and low-severity burned areas. Savanna wetland was the changed wetland-type which was most affected by the fire.

### 4.3. Correlation between the Fire and Wetland Ecological Indexes

#### 4.3.1. Dynamic Rate of Wetland Change in the Maximum Burn Range of Fire

Through the significant-level analysis of wetland changes, the change results of MNDWI were extracted from Sentinel-2 images with a spatial resolution of 10 m. Meanwhile, the changed levels of wetland ecological indexes were divided into ±50% and ±10% responding to the mean ± 1.5 standard deviation and mean ± standard deviation, respectively. For the change detection of wetlands in the burnt area, the blue area of the legend in the map represents the significant change caused by wetland fires, with a dynamic ratio of >50%. Meanwhile, the red area in the legend—mainly distributed in the unburned area—is the decreased region of seasonal water, with a change rate of ≤ 50% (Figure 8). 

In general, the change rate of MNDWI shows that the wetland hydrology factors were divided into water bodies and non-water bodies (i.e., land surface dry soil) with dynamic ratios of ≤ 50% and > 50%, respectively. Meanwhile, the dynamic ratio of wetland vegetation change led by fire ranged from 10 to 50%, showing that the fluctuation range of surface vegetation was at an insignificant level, although the local significant change was more than 50% and the overall dynamic ratio of NDSI ranged from 10 to 50%. Furthermore, the change feature of NDSI was not caused by the wildfire, but the long dry season with less water supplied for the study area.

#### 4.3.2. Wetland Change in Different Fire Severity Areas

The results of the statistical analysis of the dynamic rates of wetland ecological indexes in different fire-severity areas showed that the change of wetland ecological indexes—especially the MNDWI—in September was more significant than in August; the MNDWIs’ dynamic rates were 31.52 ± 25.75% and 34.87 ± 27.06%, respectively. Meanwhile, the dynamic rate of NDVIs in September was far less than in August, at −11.96 ± 9.37% and −22.65 ± 10.28%, respectively (Table 6). The main response of Pantanal wetland to wildfire showed a more significant variation of MNDWI and NDVI compared to NDSI.

## 5. Discussion

Spanning the period from August to September 2019, the Amazon forest fires were an unprecedented event in terms of the scale and duration of burning. The observation data of wildfires over a continuous period of 42 days and satellite imagery data, separated by 35 days, made an analysis feasible. Therefore, the analysis in this paper is unique to this particular region.

### 5.1. Accuracy Assessment

In general, the result of high-precision classification is the guarantee of determining the factual change of wetland-types, and the detection result of the multi-index-range combination shows higher calculation accuracy than a single extreme value combination [48,49,50]. Therefore, an error matrix is provided in Table 7 as an example of the classification results using the supervised classification based on the multi-dimension-feature dataset, which includes three indexes (NDVI, MNDWI, and NDBI) representing the surface vegetation information, water body information and soil information [54,55]. 

The classification result of the multi-index-range combination shows a high calculation accuracy, with the best accuracy of 80.4% (the Kappa coefficient (KC) is 0.764 in January 2019), 78.5% (KC is 0.742 in July 2019), 82.4% (0.789 in August 2019) and 81.1% (0.773 in September 2019). The extracted results of land-cover types are different, corresponding to different ecological statuses in the maximum burn area. Finally, the overall accuracy and Kappa coefficient of the classification results reached 80.6% and 0.767, respectively, and the classification results represent the actual distribution of vegetation cover-types in the study area.

Generally, compared with the original band combination, the spatial feature index (i.e., MNDWI, NDVI, NDSI) dataset increases the discrimination accuracy of wetland types significantly—especially for the area mixed by the changed information of non-wetland types—even though there is, to some extent, a limitation to the efficiency of the final results. Comprehensively, the combination based on the spatial feature index is advantageous when applied to wetland change detection.

### 5.2. Previous Works and Future Direction

The Brazilian savanna, known as the Cerrado, is a fire-prone ecosystem [56] and is also a biodiversity hotspot in which longer dry seasons lead to a higher burn frequency [57]. Wildfire is a predictable process by which the highly-dense arbor is replaced by low-density scrub. Changes in land-cover structure are associated with the recovery of some species. Fire primarily alters the region’s hydrological cycle [26]; furthermore, the water cycling is drastically diminished, with direct consequences for the local and regional climate [29]. Together with the global warming, the lengthening of the dry season is contributing to longer, hotter and drier fire seasons [18]. If these samples obtained by the VIIRS/MODIS satellite contain more information, such as detailed land cover types and dry–wet conditions, it will be of more practical significance to study the correlation between the ecological restoration duration of wetland and land type and dry–wet changes, which will also be our direction of focus in future work.

## 6. Conclusions

Using the top-of-the-atmosphere (TOA) values of blue, green, red, and NIR reflectance from Sentinel-2A 10 m imagery, a set of optimal indices—NBR, MNDWI, NDVI, and NDSI—were used to discriminate burned areas. The main objectives of this study were (1) to identify the burn scope, frequency and severity, and (2) extract the spatiotemporal change features of the wildfire and its impact on wetlands, including the change features of wetland hydrology, vegetation and soil. The final results allowed us to make the following conclusions.

(1)September 2019 was the important monitoring period when the Amazon wetland fires occurred most intensively, with a burned area of 321.1 km^2^. Meanwhile, the burn severity of the fire scars in September belonged to the low and moderate combustion level, with burnt areas of 53.42% and 46.48%, respectively.(2)Bared land (non-wetland type) and savanna wetland (wetland type) were the main land-cover types, accounting for 65.8% and 24.8% of the maximum burn range of the fire, respectively. Meanwhile, the conversion between bared land and savanna wetland was the main impact result of the fire, accounting for percentages of 68% and 88.7% of the change area of the maximum burn range, respectively.

The most important aspect of wetland ecological restoration is the determination a disturbed target. The final goal is to restore the self-balance state of wetland ecosystems, through the modification of the degraded ecological structure and function of wetland with the appropriate bioengineering technology. This study is devoted to the identification of the scope and severity of disturbance factors and their impact on the wetland ecological indexes (i.e., MNDWI, NDVI, NDSI), furthermore providing preliminary quantitative measures for these indexes. The work was conducted to rapidly determine the key restoration areas and general restoration areas after the end of the fire in the Amazon Forest and implement targeted wetland ecological restoration measures.

## Figures and Tables

**Figure 1 sensors-20-04268-f001:**
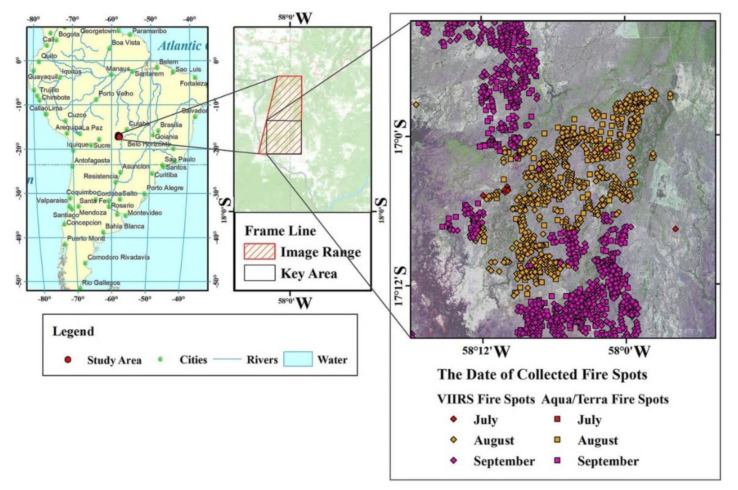
The distribution of fire sites in the Pantanal wetland. VIIRS: Visible Infrared Imaging Radiometer Suite.

**Figure 2 sensors-20-04268-f002:**
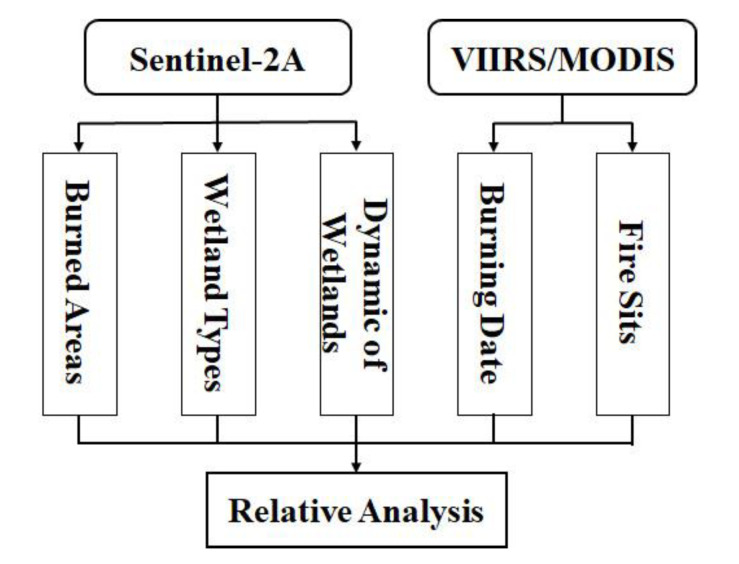
The flow chart of wetland fires in this study.

**Figure 3 sensors-20-04268-f003:**
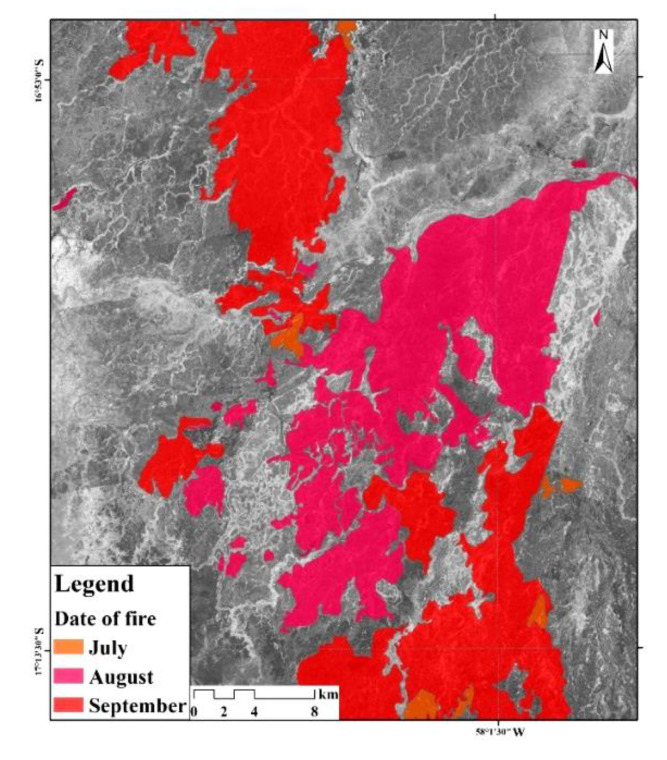
The comprehensive map of wetland fire sites, 2019.Note: Figure 3 is the superimposed map of fire sites in the study area, including the monthly burned fields in July, August and September.

**Figure 4 sensors-20-04268-f004:**
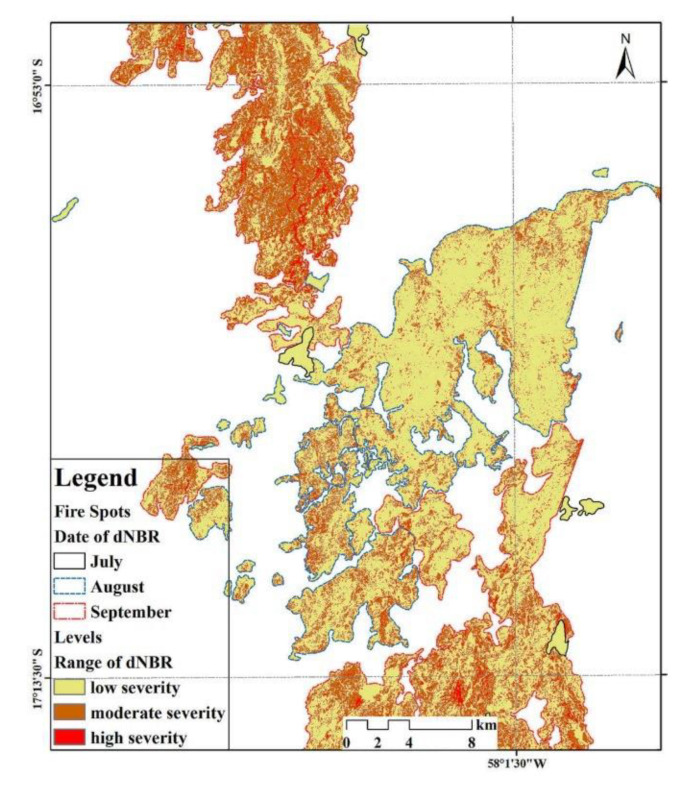
The comprehensive map of dNBR, 2019. Note: Figure 4 is the comprehensive map of the study area, including the monthly cut results (dNBR) in the burnt area between July, August and September.

**Figure 5 sensors-20-04268-f005:**
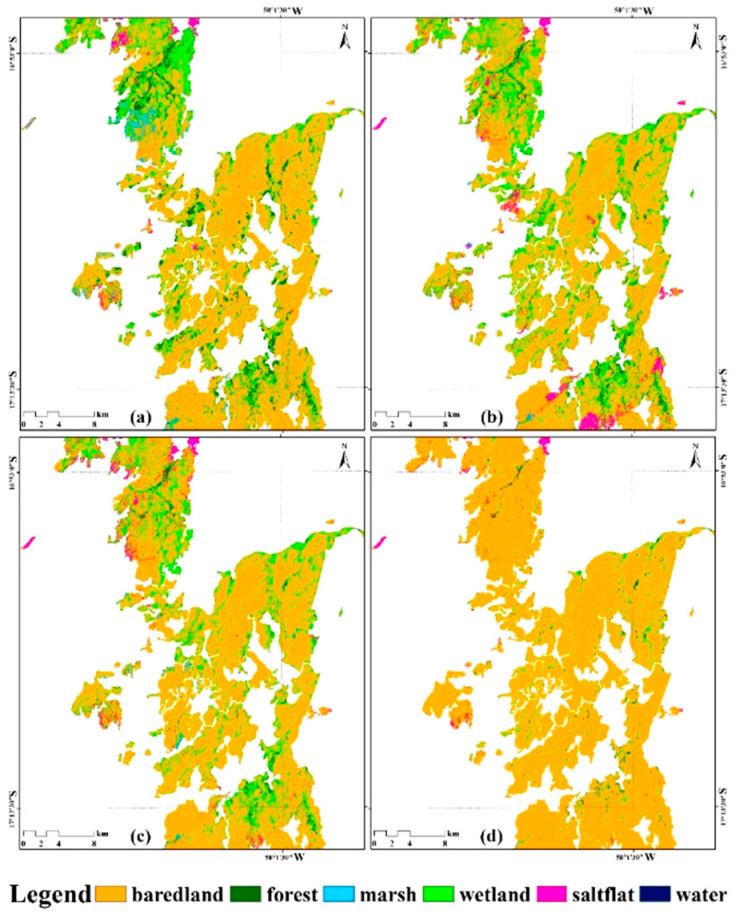
The wetland types in the maximum burn range of the fire: (**a**) the wetland types in January 2019; (**b**) the wetland types in July 2019; (**c**) the wetland types in August 2019; (**d**) the wetland types in September 2019.

**Figure 6 sensors-20-04268-f006:**
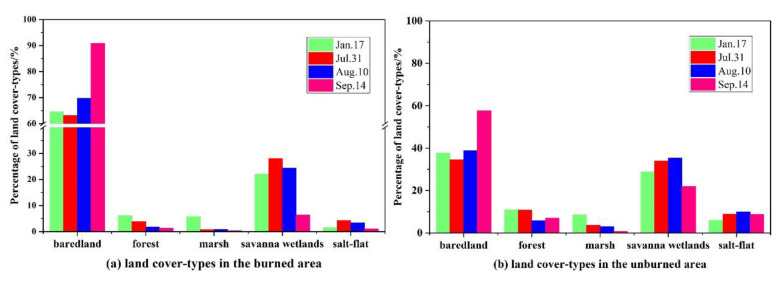
The percentage of wetland/non-wetland types: (**a**) land-cover types in the maximum burn range of the fire; (**b**) land-cover types in the unburned area.

**Figure 7 sensors-20-04268-f007:**
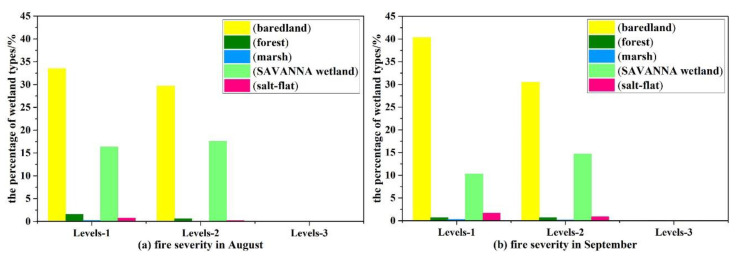
The fire severity and the percentage of wetland types: (**a**) is in August; (**b**) is in September. Note that Level-1 represents the low-severity burnt area, Level-2 represents the moderate-severity burnt area, and Level-3 represents the high-severity burnt area.

**Figure 8 sensors-20-04268-f008:**
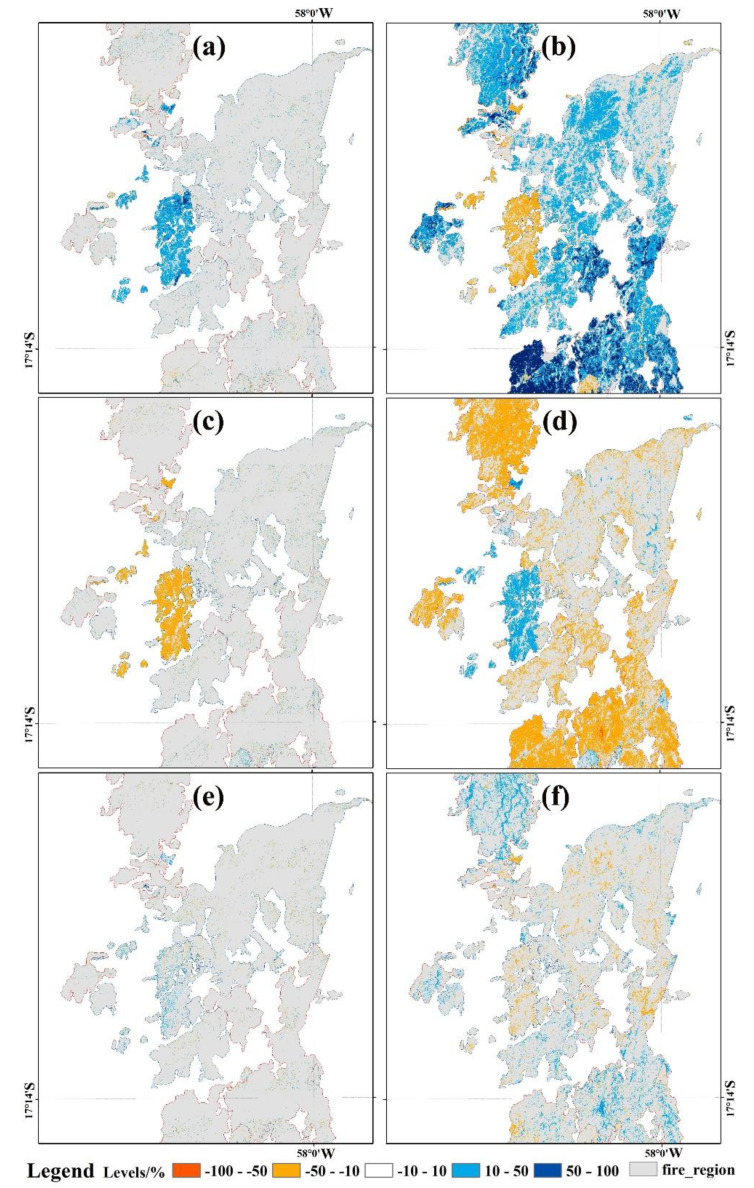
Change detection of the wetland: (**a,c,e**) in August; (**b**,**d**,**f**) in September. Note that a, b represents the dynamic rate of MNDWI, c, d represents the dynamic rate of NDVI, and e, f represents the dynamic rate of NDSI.

**Table 1 sensors-20-04268-t001:** The fire spot distribution in the Pantanal wetland from August to September, 2019. MODIS: Moderate Resolution Imaging Spectroradiometer.

Sensor	Fire Spots	Temporal Range
VIIRS	1077	23 July 2019–30 September 2019
MODIS	238	23 July 2019–30 September 2019

**Table 2 sensors-20-04268-t002:** The collection of the remote sensing data.

Sensor	Date	Season	Track Number
Sentinal-2A	17 January 2019	Wet season	213/067
	31 July 2019	Dry season	213/067
	10 August 2019		
	14 September 2019		

**Table 3 sensors-20-04268-t003:** The burned area and fire severity in the study area. NBR: Normalized Burn Ratio; dNBR: difference of Normalized Burn Ratio.

Month	Area	NBR	dNBR
	/km^2^	Mean ± Std	Mean ± Std
July 2019	13.2	−0.134 ± 0.109	0.003 ± 0.034
August 2019	264.7	−0.141 ± 0.159	0.203 ± 0.122
September 2019	321.1	−0.162 ± 0.146	0.028 ± 0.037

**Table 4 sensors-20-04268-t004:** Fire severity and conversion of wetland types. Unit: %.

	Fire	Marsh	Savanna Wetland	Salt Flat
Date	Severity	Transfer-in/-out	Transfer-in/-out	Transfer-in/-out
August 2019	Level-1	2.89/0.7	26.75/63.14	5.13/4.29
	Level-2/3	2.41/0.44	22.09/73.14	2.73/1.48
September 2019	Level-1	4.12/1.99	2.20/72.13	0.66/21.35
	Level-2/3	0.96/0.6	0.39/88.45	0.09/6.26

**Table 5 sensors-20-04268-t005:** Fire severity and the conversion of non-wetland types. Note: Statistical change pixels include those of the wetland and non-wetland types in the study area.

	Fire	Bared Land	Forest	Change Pixels
Date	Severity	Transfer-in/-out (%)	Transfer-in/-out (%)	Total
August 2019	Level-1	64.73/20.24	0.49/11.64	177,438
	Level-2/3	72.54/17.18	0.22/7.75	132,675
September 2019	Level-1	89.43/0.87	3.59/3.67	261,478
	Level-2/3	97.97/0.14	0.58/4.55	615,165

**Table 6 sensors-20-04268-t006:** The dynamic rate of ecological indexes and fire severity. Unit: %. dMNDWI: difference in Modified Normal Differential Water Index; dNVDI: difference in Normalized Vegetation Difference Index; dNDSI: difference in Normalized Difference Soil Index.

	Fire	dMNDWI	dNDVI	dNDSI
Date	Severity	Mean/SD	Mean/SD	Mean/SD
August 2019	Level-1	3.24/10.6	(−1.37)/9.3	(−0.156)/5.56
	Level-2/3	7.12/14.75	(−5.3)/14.05	0.57/7.06
September 2019	Level-1	31.52/28.75	(−11.96)/9.37	(−0.87)/8.57
	Level-2/3	34.87/27.06	(−22.65)/10.28	8.28/9.36

**Table 7 sensors-20-04268-t007:** The error matrix of classification results. Note: BL, bared land, F, forest, W, water bodies, M, marsh, OWSF, open wood savanna–flood, SF, salt flat, PA and UA are the producer’s and user’s accuracies, respectively.

Accuracy Assessment Results Using Support Vector Machine, in Jan. 2019								
Class	BL	F	W	M	OWSF	SF	UA	PA
BL	73	0	0	0	5	6	86.9	74.49
F	8	61	0	9	6	0	72.62	70.11
W	0	3	76	0	0	0	96.2	84.44
M	0	13	5	68	5	0	74.73	88.31
OWSF	17	10	0	0	77	9	68.14	82.8
SF	0	0	9	0	0	75	89.29	83.33
Accuracy assessment results using support vector machine, in Jul. 2019								
BL	76	0	0	0	4	7	87.36	77.55
F	0	65	0	11	12	0	73.86	74.71
W	0	4	74	0	0	0	94.87	83.15
M	0	10	15	67	0	0	72.83	85.9
OWSF	22	8	0	0	69	14	61.06	74.19
SF	0	0	0	0	8	69	89.61	76.67
Accuracy assessment results using support vector machine, in Aug. 2019								
BL	79	0	0	2	8	0	88.76	80.61
F	0	58	0	8	6	0	80.56	66.67
W	0	0	70	2	0	0	97.22	87.5
M	0	15	7	75	0	0	77.32	86.21
OWSF	19	14	0	0	72	3	66.67	77.42
SF	0	0	3	0	7	87	89.69	96.67
Accuracy assessment results using support vector machine, in Sep. 2019								
BL	73	10	0	0	14	8	69.52	74.49
F	0	59	0	5	5	0	85.51	64.84
W	0	4	71	0	0	0	94.67	89.87
M	3	10	8	79	0	0	79.00	94.05
OWSF	12	8	0	0	74	5	74.75	79.57
SF	10	0	0	0	0	77	88.51	85.56

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
