# Peer review of "Wetland Fire Scar Monitoring and Its Response to Changes of the Pantanal Wetland"

_sensors, 2020, doi:10.3390/s20154268_

Round 1

Reviewer 1 Report

It is appreciated that the authors revised this manuscript quickly.  However, several of the recommendations in the original review were ignored.  There are improvements and the manuscript contains considerable useful science on a topic of interest to many scientists and decision makers.

Accuracy assessment is important in almost every remote sensing assessment.  The authors need to fully clarify how they conducted such an assessment.  How many samples were selected for calibration and validation and from what sources.  A complete error matrix should be included specifying the classes and standard user and producer accuracies.  The discussion near line 193 is confusing.  What is the function of training (calibration) samples for manual interpretation?   It also appears that the accuracy assessment was for four land cover types and not the areas of burn. 

There are some editorial suggestions for the authors, some of which are repeated from the earlier review:

  1. Typically an abstract contains specific, often numerical, results such as areas of burn, OA, etc.
  2. Line 37, affect for effect.
  3. Line 67, birds and beetles.
  4. Line 82, were for was.
  5. Line 91, RS spell out first use but really not necessary, should be lower case.
  6. Insert tables and figures after citation in text.
  7. The use of personal pronouns (we) is not typical in scientific text but a journal editorial decision.
  8. M or meter?
  9. Line 150, cartographer? Map or cartography?
  10. Line, 157, delete And.
  11. Line 158, from for form.
  12. Line 163, forming? Mapping?
  13. Line 173, Eq-5?
  14. Line 175, elsewhere?
  15. Line 179, MIR or SWIR? Be consistent.
  16. Line 186, situate?
  17. Line 201, move and.
  18. Line 215, August,
  19. Line 218, incomplete sentence.
  20. Line 244 and elsewhere. Bared?  Bare perhaps?
  21. Line 296, main for mainly.
  22. As previously, the references are not in consistent format. Article titles are in both upper and lower case, journal titles are abbreviated and not, etc.

This is competent research.  The manuscript could still be shorted and needs some clarifications especially on thematic accuracy assessment.   There is also a need for more editorial review.

Author Response

Response to Reviewer 1 Comments

Response 1. "The authors need to fully clarify how they conducted such an assessment. How many samples were selected for calibration and validation and from what sources."

Point 1: In this revision process, the newly added contents are the selection and training of sampling points in the study area. The details are as follows.

……3.2.3 Wetland types and accuracy evaluation

Data of land cover are referenced from the global surface coverage remote sensing data products (GlobeLand30). The main wetland types of remote sensing classification in the study area include constant water bodies, marsh (Swampy Grassland), Savanna wetland (OWSF, Open Wood Savanna-flood). The Sentinel-2A multispectral images in Jan. Jul. Aug. and Sep. 2019 with a spatial resolution of 10 meters are utilized as the main data source combined with the auxiliary data, which mainly refer to the Google Earth high-resolution images. The supervised image classification will classify the image based on the training sets provided by the user based on his field knowledge. The training data given by the user guides the software as to what types of pixels are to be selected for certain land cover type.

To evaluate the applicability of these data in the study area, 535 points in the core areas are randomly collected and extracted by computer, and then they are superimposed with a high-resolution satellite image from Google Earth® for the accuracy test. Finally, the selected samples are randomly divided into two parts: 70% of the samples which are used for the classification, and 30% of the samples which are used for the accuracy evaluation. The main quantitative evaluation of the results includes the overall accuracy (OA), Kappa coefficient (KC) [40].

Response 2. "A complete error matrix should be included specifying the classes and standard user and producer accuracies."

Point 2: In this revision process, the contents of the "5.1 Accuracy analysis "section are rewritten. The details are as following that:

……5.1 Accuracy Assessment

In general, the result of high precision classification is the guarantee to determine the factual change of wetland-types, and the detection result of multi-index-range combination shows higher calculation accuracy than a single extreme value combination [48, 49, and 50]. Therefore, an error matrix is provided in Table 7 as an example of the classification results using the supervised image classification based on the multi-dimension-feature dataset, which included three indexes (NDVI, MNDWI, and NDBI) representing the surface vegetation information, water body information, and soil information [53, 54].

Table 7. The error matrix of classification results

Accuracy assessment results using support vector machine, in Jan. 2019

Class

BL

F

W

M

OWSF

SF

UA

PA

BL

73

0

0

0

5

6

86.9

74.49

F

8

61

0

9

6

0

72.62

70.11

W

0

3

76

0

0

0

96.2

84.44

M

0

13

5

68

5

0

74.73

88.31

OWSF

17

10

0

0

77

9

68.14

82.8

SF

0

0

9

0

0

75

89.29

83.33

Accuracy assessment results using support vector machine, in Jul. 2019

BL

76

0

0

0

4

7

87.36

77.55

F

0

65

0

11

12

0

73.86

74.71

W

0

4

74

0

0

0

94.87

83.15

M

0

10

15

67

0

0

72.83

85.9

OWSF

22

8

0

0

69

14

61.06

74.19

SF

0

0

0

0

8

69

89.61

76.67

Accuracy assessment results using support vector machine, in Aug. 2019

BL

79

0

0

2

8

0

88.76

80.61

F

0

58

0

8

6

0

80.56

66.67

W

0

0

70

2

0

0

97.22

87.5

M

0

15

7

75

0

0

77.32

86.21

OWSF

19

14

0

0

72

3

66.67

77.42

SF

0

0

3

0

7

87

89.69

96.67

Accuracy assessment results using support vector machine, in Sep. 2019

BL

73

10

0

0

14

8

69.52

74.49

F

0

59

0

5

5

0

85.51

64.84

W

0

4

71

0

0

0

94.67

89.87

M

3

10

8

79

0

0

79.00

94.05

OWSF

12

8

0

0

74

5

74.75

79.57

SF

10

0

0

0

0

77

88.51

85.56

Note: BL, bared land, F, forest, W, water bodies, M, marsh, OWSF, open wood Savanna-flood, SF, salt-flat.

The classification result of multi-index-range combination shows higher calculation accuracy with the best accuracy of 80.4% (KC is 0.764 in Jan. 2019), 78.5% (0.742, Jul. 2019), 82.4% (0.789, Aug. 2019), and 81.1% (0.773, Sep. 2019). The extracted result of land-cover types is different corresponding to different ecological status in the max-burnt-area. Finally, the overall accuracy and Kappa coefficient of the classification results reached 80.6% and 0.767, respectively, and the classification results can represent the actual distribution of vegetation cover-types in the study area.

Response 3. "The discussion near line 193 is confusing. What is the function of training (calibration) samples for manual interpretation? "

Point 3: The discussion near line 193 has been rewritten. See the Point 1 for details.

Response 4. "There are some editorial suggestions for the authors, some of which are repeated from the earlier review: 1.Typically an abstract contains specific, often numerical, results such as areas of burn, OA, etc."

Point 4: (ABSTRACT) Fire is an important disturbance factor......Finally, the overall accuracy and Kappa coefficient of the extracted results of wetland types reached 80.6% and 0.767, respectively...... Meanwhile, fire scars belong to the moderate- and low-severity burnt areas, with the max-burn-area of 599km2.

Response 5. "7. The use of personal pronouns (we) is not typical in scientific text but a journal editorial decision."

Point 5: "We selected the Sentinel-2A multispectral data (2019) to monitor......"

Point-5.1: The Sentinel-2A multispectral data were selected to monitor the actual process and specific situation of fire occurring in the study area (Table 2).

"We conducted analysis processes which include the calculation of NBR......"

Point-5.2: To investigate whether the Sentinel-2A data can capture the short-term burning events and seasonal dynamics for wetlands, the analysis processes include the calculation of NBR, and the Dynamic Ratio, as described below with the Sentinel-2A datasets (Figure 2).

"We use the mean value of ecological indexes of two images to represent the constant......"

Point-5.3: The mean value of ecological indexes of two images is used to represent the constant status at the same location......

"We manually interpreted and extracted the Pantanal wetland burnt area in the Sentinel-2A......"

Point-5.4: the Pantanal wetland burnt or burning fields were manually interpreted and extracted in the Sentinel-2A images from July to September 2019.

"We obtained the stacking layer of wetland fire severity......"

Point-5.5: The stacking layer of wetland fire severity was obtained using the NBR calculated from Sentinel-2A images......

"We conducted the zonal statistics of wetland types in the burnt area......"

Point-5.6: The zonal statistics of wetland types in the burnt area were conducted based on the ArcGIS platform.

Response 6. "15. Line 179, MIR or SWIR? Be consistent."

Point 6:We removed the notion of "MIR", and replaced for SWIR1.

NBR = (NIR-SWIR2)/ (NIR+SWIR2).................. (1)

Similar to the formula of NDVI, the NBR uses the near infrared (NIR) and the short-wave infrared 2 (SWIR2) wavelengths which were corresponding to band 8 (0.76-0.90μm) and band 12 (2.07-2.32μm), respectively.

MNDWI= (Green- SWIR1)/ (Green + SWIR1).................. (3)

The normalized difference soil index (NDSI) is also known as conventional soil index. The ratio is based on the near infrared (NIR) and the short-wave infrared 1 (SWIR1, 1.57-1.65μm) bands of remote sensing image data. The negative value of NDSI represents water body’s areas [49].

Response 7. "22. As previously, the references are not in consistent format. Article titles are in both upper and lower case, journal titles are abbreviated and not, etc."

Point 7: All grammar problem and reference formats have been revised point by point. Because time is short, we asked the English polishing company (L&L Geospatial Information technologies, LLC) to make a 24-hour urgent deal of the revised manuscript.

Reviewer 2 Report

This study monitored the Amazon Forest fires in 2019 with Sentinel-2A data, addressing an important topic, and demonstrating a nice application of the Sentinel-2A data. My main critic of the study is that, when the remote sensing data from July to September can be used to map the burn scar, they can not be used to evaluate the ecological impact much, as they show no information of the wet season following the fire, which starts in December.  And overall, the structuring and writing of the manuscript need to be improved. Some specific comments are followed.  

There seems to be a formatting issue. Why some paragraph appears to be in grey shading?

Line 29-42. Statements are contradicting each other and confusing.

It has been confirmed by Brookman-amissah et al. [9] that … fires tend to increase species diversity, while at the end of the dry season, and they will decrease the species diversity in the long-term. That is why fire … to promote biodiversity conservation, especially in the African and Australian savannas [11]

Overall, in the introduction, the discussion on fire impact on wetland ecology and review of the literature is not well organized and very confusing. Also, it would be good to provide some information about the Amazon fire event in 2019 itself.

Line 91 what is ‘RS’

Line 212,  ‘we  manually  interpreted  and  extracted ….Finally, a total of 30 burnt or burning fields were divided from the overfired area of the Pantanal  wetland park, among which 7, 12, and 11 fire fields were existed in July, August  and September, respectively.’  The method needs to be explained better.

I am confused about Table 3.  How is the dNBR calculated, what is the reference image?

What is being shown in figure 4, the dNBR between August and July? As mentioned in the text. Or dNBR between September and July, as mentioned in the caption.

Table 4, what land cover time would have been transfer-in as Savanna Wetland in August?

Line 317, ‘Meanwhile,  the  dynamic  ratio  of  wetland  vegetation  change  led  by  fire

ranges from 10% to 50%, denoting that wildfires cannot convert wetland types.’ I don’t not understand this sentence.

Line 329 ‘The main response of Pantanal wetland to wildfire is the change of ecological factors such as surface water and terrestrial vegetation.’ What kind of responses is this sentence referring to? What can be inferred from the MNDWI and NDVI to the changes in the water and vegetation?

In the conclusion section, the contribution of this study in ecological restoration is not well explained.  

Author Response

Response to Reviewer 2 Comments

Response 1. "……they cannot be used to evaluate the ecological impact much, as they show no information of the wet season following the fire, which starts in December. And overall, the structuring and writing of the manuscript need to be improved. "

Point-1.1: In this revision process, we emphasize the change of wetland types and weaken the ecological impart of fire on wetland. After this modification, the focus of this manuscript is adjusted to the change of wetland types in the burnt area.

Example:

……The research enriches the studying cases of the influence of wildfire as the main disturbance factor on the change of wetland types, provides scientific reference for the restoration and sustainable development of global wetland ecosystems.

Point-1.2: All grammar problem and reference formats have been revised point by point. Because time is short, we asked the English polishing company (L&L Geospatial Information technologies, LLC) to make a 24-hour urgent deal of the revised manuscript.

Response 2. "Line 29-42. Statements are contradicting each other and confusing.

It has been confirmed by Brookman-amissah et al. that … fires tend to increase species diversity, while at the end of the dry season, and they will decrease the species diversity in the long-term. That is why fire … to promote biodiversity conservation, especially in the African and Australian savannas … in the introduction, the discussion on fire impact on wetland ecology and review of the literature is not well organized and very confusing. Also, it would be good to provide some information about the Amazon fire event in 2019 itself."

Point 2: In the discussion of the effects of fire, a concrete example which provide some information about the Amazon fire event in 2019, is used to instead of other explanations.

Most wetland fires refer to the naturally occurring wildfires, which is considered one of the important environmental disturbing factors [1-4]. In general, wildfires have negative environmental effects can occur such as global warming, land fertility, ecological diversity, land-cover types, and people's property, but also positive impacts on vegetation regeneration and nutrient recycling [5-8]. With the increasingly frequent wildfires in South America during the last 15 years, the grasslands and savannas present high mortality. Meanwhile, the fire spread is greatly accelerated through the continuous grass layer that characterizes grasslands and savannas leading to more open vegetation types [9]. That is why fire was been recognized as a common tool to remove biomass from newly cleared fields [10] and a management tool to promote biodiversity conservation [11].

Response 3. "Line 91 what is ‘RS’"

Point 3: 'RS' belongs to the non-standard utility of vocabulary has been banned.

Response 4. "Line 212, ‘we manually interpreted and extracted ….Finally, a total of 30 burnt or burning fields were divided from the overfired area of the Pantanal wetland park, among which 7, 12, and 11 fire fields were existed in July, August and September, respectively.’ The method needs to be explained better."

Point 4: The production process of comprehensive diagram is divided into the following three steps described in detail.

Based on the premise of ensuring the continuity and integrity of the spatial distribution of fires, the Pantanal wetland burnt or burning fields are manually interpreted and extracted in July, August, and September by computer, respectively. Then, a total of 30 burnt or burning fields are divided from the overfired area of the Pantanal wetland park, among which 7, 12, and 11 fire fields are existed in July, August, and September, respectively. Finally, the extracted results of each month are superimposed to obtain the comprehensive map from July to September 2019.

Response 5. "How is the dNBR calculated, what is the reference image?"

Point 5: Firstly, the combustion index (NBR) for January, July, August, and September are calculated based on the Sentinel-2A images, respectively. Then, the monthly difference of combustion index (dNBR) are calculated and cut using the fire region vector data of the corresponding month to obtain the cut results in the burnt area. Finally, the monthly cut results are superimposed to obtain the comprehensive map of the main months from July to September 2019.

Response 6. "What is being shown in figure 4, the dNBR between August and July? As mentioned in the text. Or dNBR between September and July, as mentioned in the caption."

Point 6: the figure 4 is the comprehensive map in the study area, including the monthly cut results (dNBR) in the burnt area in July, August, and September, respectively.

Response 7. "Table 4, what land cover time would have been transfer-in as Savanna Wetland in August?"

Point 7:

The conversion matrix of land-cover types from July-August unit: km2

ch-78

bared land

forest

water

marsh

Savanna wetland

salt-flat

bared land

357.40

0.04

1.04

0.13

1.48

6.95

forest

0.58

9.97

0.01

0.37

12.50

0.01

water

0.98

0.00

0.78

0.25

0.01

0.00

marsh

0.54

0.05

0.10

0.99

0.55

0.01

Savanna wetland

43.46

0.32

0.48

0.68

120.33

1.85

salt-flat

13.67

0.01

0.01

0.09

0.70

10.88

In August, the change characteristics of savanna wetland are mainly manifested as large-scale transfer-out, with the change area of 43.46 km2. Part of the open-savanna-forest in the burnt area is transformed into savanna wetland, with the change area of 12.5 km2.

Response 8. "Line 317, 'Meanwhile, the dynamic ratio of wetland vegetation change led by fire ranges from 10% to 50%, denoting that wildfires cannot convert wetland types.' I don't not understand this sentence."

Point 8: ......, denoting that the fluctuation range of surface vegetation belongs to the insignificant level.

Response 9. "Line 329 ‘The main response of Pantanal wetland to wildfire is the change of ecological factors such as surface water and terrestrial vegetation.’ What kind of responses is this sentence referring to? What can be inferred from the MNDWI and NDVI to the changes in the water and vegetation?"

Point 9: The main response of Pantanal wetland to wildfire is the variation of MNDWI and NDVI.

Response 10. "In the conclusion section, the contribution of this study in ecological restoration is not well explained."

Point 10: the newly added contents are as following that:

To determine a disturbed target is the most important thing for the wetland ecological restoration. The finally achievement is restored the self-balance state of wetland ecosystem, through the modification of the degraded ecological structure and function of wetland with the appropriate bioengineering technology. The study is devoted to identify the scope and severity of disturbance factors and their impact on the wetland ecological factors (i.e. MNDWI, NDVI, NDSI), furthermore, preliminary quantitative measure to them. The works are conducted to determine rapidly the key restoration areas and general restoration areas after the end of the fire, and implement the targeted wetland ecological restoration measures.

Reviewer 3 Report

Using remote sensing method to timely assess the fire impact on wetlands is necessary to wetland conservation and management. The authors present a good sample in Amazon wetlands, which is one of the most important tropical forest wetlands on Earth.

The author could present the status of ecological environment in Amazon forest wetlands which fire often happens and provide the innovations of this research.

The approach used in this research could be discussed more details in the discussion section from the strength or weakness.

The manuscript should be polished by English-native experts. There are many grammar errors in its current version.

Some of the figures should be redrawn. For example, the figure1. The legend is too vague to tell from different dates. Regarding to figure 4, the legend should be altered to show which level of the severity. The legend of bareland and salt-flat seem to be identical in figure 7.

Author Response

Response to Reviewer 3 Comments

Response 1. “The approach used in this research could be discussed more details in the discussion section from the strength or weakness.”

Point 1:Generally, compared with the original band combination, the spatial feature index (i.e. MNDWI, NDVI, NDSI) dataset increases the discrimination accuracy of wetland types significantly, especially for the area mixed by the changed information of non-wetland types, even though there is, to some extent, a limitation to the efficiency of the final results. Comprehensively, the combination based on the spatial feature index is advantageous when applied to the wetland change detection.

Response 2.”The manuscript should be polished by English-native experts. There are many grammar errors in its current version.”

Point 2:All grammar problem and reference formats have been revised point by point. Because time is short, we asked the English polishing company (L&L Geospatial Information technologies, LLC) to make a 24-hour urgent deal of the revised manuscript.

Response 3.”Some of the figures should be redrawn. For example, the figure1. The legend is too vague to tell from different dates. Regarding to figure 4, the legend should be altered to show which level of the severity. The legend of bared land and salt-flat seem to be identical in figure 7.”

Point 3:In this revision process, the newly modified contents are as following that:

Point-3.1:

Figure 1. The distribution of fire sites in the Pantanal wetland

Point-3.2: Firstly, the combustion index (NBR) for January, July, August, and September are calculated based on the Sentinel-2A images, respectively. Then, the monthly difference of combustion index (dNBR) are calculated and cut using the fire region vector data of each month to obtain the cut results in the burnt area. Finally, the monthly cut results are superimposed to obtain the comprehensive map of the main months from July to September 2019.

Figure 4. The comprehensive map of dNBR, 2019

Note: the figure 4 is the comprehensive map in the study area, including the monthly cut results (dNBR) in the burnt area between July, August, and September.

Point-3.3:

Figure 7. The fire severity and the percentage of wetland types: Figure 7-(a) is in August; Figure 7-(b) is in September. Note that: the Level-1 represents the low-severity burnt area, the Level-2 represents the moderate-severity burnt area, and the Level-3 represents the high-severity burnt area.

Round 2

Author Response

  1. The research enriches the studying cases of the influence of wildfire as the main disturbance factor on the change of wetland types.

Reply-1: The case enriches the studying of the impact of wildfire as the main disturbance factor on the change of wetland types.

  1. The fire spread is greatly accelerated through the continuous grass layer that characterizes grasslands and savannas leading to more open vegetation types.

Reply-2: The fire spread is greatly accelerated that lead to more open land-cover types, through the continuous grass layer characterized by the grasslands and savannas.

  1. Especially, the slash-and-burn agriculture caused from the human activities has the novel disturbances to the land-cover types in Amazon, with the mainly increasing sources of ignition and the more flammable landscape structure.

Reply-3: Especially, the slash-and-burn agriculture caused from the human activities has the new disturbance to the land-cover types……

  1. The study on the post-fire impact on the ecological factors include the following aspects……

Reply-4: The study on the post-fire impact on the ecological factors includes the following aspects……

  1. The post-wildfire hydrologic response, however, does not always decrease monotonically with increasing time since fire.

Reply-5: However, the change of hydrological response after fire does not always decrease gradually and linearly with time.

  1. Combined with the post-fire climate change, wildfire, as the main disturbances, will determine the future land-cover type and whether these cover-types can sustain critical habitat types and ecosystem services over the long term, or will enter a common period of degradation.

Reply-6: Combined with the post-fire climate change, wildfire, as the main disturbance, will determine the future land-cover types and whether these land-cover types can sustain critical habitat types and ecosystem services over the long term, or will enter a common period of degradation.

  1. The contribution of this research includes the analysis between the fire and the change of wetland types can enrich the research cases of the influence of wildfire as the main disturbance factor on the change of wetland types.

Reply-7: The analysis of the relationship between the fire and the change of wetland types can enrich the research of the wildfire influence as the main disturbance factor on the change of wetland types.

  1. The Pantanal wetland is the largest tropical inland plain wetland globally, is mainly located in the South Mato Grosso.

Reply-8: The Pantanal Wetland, as the largest tropical inland plain wetland in the world, is mainly located in the South Mato Grosso.

  1. The Pantanal wetland is the distribution region of the highest diversity of aquatic plants and is the largest freshwater wetland ecosystem.

Reply-9: The Pantanal wetland has the highest diversity of aquatic plants and is the largest freshwater wetland ecosystem in the world.

  1. Wildfires leading to wetland type's changing is not only an important disturbance factor for ecosystems, but also a significant influence factor of global atmosphere quality.

Reply-10: The wildfire which led to wetland type's change is not only an important disturbance factor for ecosystems, but also a significant influence factor of global atmosphere quality.

  1. The Sentinel-2A multispectral images in Jan. Jul. Aug. and Sep. 2019 with a spatial resolution of 10 meters are utilized as the main data source combined with the auxiliary data......

Reply-11: The Sentinel-2A multispectral images in January, July, August, and September 2019 with a spatial resolution of 10 meters are utilized as the main data source for the classification combined with the auxiliary data for the accuracy evaluation......

  1. The supervised classification is used to classify the image based on the training sets provided by the user based on his field knowledge.

Reply-12: The supervised classification is used to classify the image based on the training sets provided by the mapper's professional knowledge and the field survey.

  1. To evaluate the applicability of these data in the study area, 535 points in the core areas are randomly collected and extracted by computer, and then they are superimposed with a high- resolution satellite image from Google Earth® for the accuracy test. Furthermore, the selected samples are randomly divided into two parts: 70% of the samples which are used for the classification, and 30% of the samples which are used for the accuracy evaluation. The main quantitative evaluation of the results includes the overall accuracy (OA), Kappa coefficient (KC).

Reply-13: To evaluate the applicability of the Sentinel-2A images, 1784 points in the core areas are randomly collected and extracted by computer, and then they are superimposed with a high- resolution satellite image from Google Earth® for the classification and the accuracy evaluation. Furthermore, the selected samples are randomly divided into two parts: 70% of the samples which are used for the classification, and 30% of the samples which are used for the accuracy evaluation. The main quantitative evaluation of the results includes the overall accuracy (OA), Kappa coefficient (KC).

  1. The Pantanal wetland burnt or burning fields in July, August, and September are manually interpreted and extracted by computer, respectively.

Reply-14: The Pantanal wetland's burnt and burning fields in July, August, and September are manually interpreted and extracted by mapper combined with the computer, respectively.

  1. Finally, the monthly cut results are superimposed to obtain the comprehensive map of the main months from July to September 2019.

Reply-15: Finally, the monthly cut results are superimposed to obtain the comprehensive map (Figure 4) of the main months from July to September 2019.

  1. Because the dNBR is the difference-calculation result, we can see the overlay results including that of the fire-region in August and September.

Reply-16: the difference calculation results provided by this paper are the superimposed map of the fire-region in August and September.

  1. It can be seen that the main land cover-type in the maximum-burn-range was Savanna wetland which was the only wetland type impacted by the fire......

Reply-17: The main land cover-type in the maximum-burn-range was Savanna wetland which was the main wetland type impacted by the fire......

  1. First of all, it should be emphasized that the percentage of Savanna wetland in the burnt area continued to decrease after entering the dry season, so did the actual affected area of water.

Reply-18: First of all, the percentage of Savanna wetland in the burnt area continued to decrease after entering the dry season, so did the actual water area affected.

  1. Through the significant-level analysis of wetland changes, the wetland ecological factors change results are extracted from Sentinel-2 images......

Reply-19: Through the significant-level analysis of wetland changes, the change results of MNDWI are extracted from Sentinel-2 images......

  1. ......the Blue frame of the legend ......the Red frame in the legend......

Reply-20: ......the Blue area of the legend ......the Red area in the legend......

  1. The main response of Pantanal wetland to wildfire is the variation of MNDWI and NDVI.

Reply-21: The main response of Pantanal wetland to wildfire showed more significant variation of MNDWI and NDVI comparing to NDSI.

  1. ......again, the method for the accuracy assessment is not clear to me. How many and what are the sampling points are used for the accuracy assessment?

Reply-22: The modification in the section is combined with the reply-11, 12, and 13.

Finally, 535 sample points are used for the accuracy evaluation of the classification results, accounting for 30% of the total sample size.

  1. The method should be mentioned in the method section……

Reply-23: The classification algorithm used by this research is the support vector machine (SVM), because the SVM algorithm can achieve better classification accuracy based on the small samples selected from the study area [58].

Cortes, C. and Vapnik, V. Support Vector Machine. Machine Learning, 1995.20, 273-297. DOI: 10.1007/BF00994018.

  1. September 2019 is the main monitoring period when the Amazon wetland fires occurred intensively, with the burnt area of 321.1km2.

Reply-24: September 2019 was the important monitoring period when the Amazon wetland fires intensively occurred, with the burnt area of 321.1km2.

  1. Bared land (non-wetland type) and Savanna wetland (wetland type) are the main distribution-types, accounting for 65.8% and 24.8%.

Reply-25: Bared land (non-wetland type) and Savanna wetland (wetland type) are the main land-covers types, accounting for 65.8% and 24.8%.

  1. To determine a disturbed target is the most important thing for the wetland ecological restoration.

Reply-26: The most important thing for the wetland ecological restoration is to determine a disturbed target.

This manuscript is a resubmission of an earlier submission. The following is a list of the peer review reports and author responses from that submission.

Round 1

Reviewer 1 Report

I appreciated the opportunity to review the manuscript “Wetland fire scar monitoring and its respond [sic] to change of the Pantanal wetland.” The topic of the recent wildfires in the Amazon region is of broad scientific interest, and of concern due to the potential for land cover change, disturbances to species and hydrology regionally, and even due to concern over carbon emissions globally.

The Introduction contains several assertions that do not appear substantiated by the references chosen by the authors as citations, and indeed many of these would be disputed broadly by researchers who specialize in wetlands and disturbance ecology. In general the first three paragraphs of the manuscript contain several such instances, and altogether do not serve as a proper background on the problem of wildland fires in wetland ecosystems that the authors wish to present as a larger context for their work.

The significance of this work appears to be in the measurement of burned area in the Pantanal and the time required for wetland vegetation recovery following fires of multiple severity categories. It appears that a bit more extrapolation is undertaken to predict ecological change than can be justified by the methods used in the manuscript.

My recommendation is that the authors recast this paper as a measurement of burn area and spectral recovery following the fire events of 2019, pare down the findings to include only relevant results, and avoid ecological interpretations. These actions might result in a suitable paper to this journal, but would effectively constitute a different submission.

Reviewer 2 Report

V

The employment of remote sensing for various aspects of fires including prediction, real-time monitoring, assessment and recovery is well developed as the authors correctly discuss.  Also as the authors discuss, fires are increasing globally.  As an aside however, fires are not necessarily always negative.  There are a number of ecosystems which are a result of fires and intentional burns are a strategy to maintain those ecosystems.  The authors provide a very useful literature review on remote sensing and fires.  One of the liabilities of this manuscript for a remote sensing journal is the lack of accuracy assessment in the results.

This is a complex manuscript.  The authors are to be congratulated on an extensive amount of data collection and especially analysis.  However, that is also a limitation as the manuscript is lengthy and the manuscript would benefit from being more focused.  A reduction in the number of tables and figures and discussion would make the manuscript more focused and thus more useful to the community.

The text requires editorial review as there are many long sentences and questionable word choices.  There is an extensive set of appropriate references but not in consistent or complete format.  

As in almost every manuscript, there are editorial suggestions for consideration by the authors, several selected ones follow:

  1. Line 35, occurance…causes.
  2. The headings do not seem to be in consistent format.
  3. Tables and figures should be cited in the text and then inserted after citation.
  4. The captions for tables and figures are not in consistent format.
  5. Line 56, wetlands.
  6. Line 66, delete ‘by humans’.
  7. Lines 78-79, delete “To estimate”.
  8. Line 87, and (3), ‘directly impacted’
  9. The employment of personal pronouns (we, us) is not typical in scientific text.
  10. Line 88, quantitatively.
  11. Lines 101 and 107. The Park size appears very small relative to the entire wetland.
  12. 1. The scale bar apparently is for the map of So. Am. which is not needed.  
  13. What is the size of the study area?
  14. Table 1, MODIS for Aqua/Terra?
  15. Line 139, data are plural.
  16. Figure 2, Sentinel.
  17. Line 172, for a cartographer, large-scale is a small area. Regional or large area?
  18. Line 195, spell out NIR.
  19. Line 202 and elsewhere, bare.
  20. Line 207, short wave infrared?
  21. Line 271, months 7,8 and 9? This figure and table 4 do not seem necessary.
  22. Line 304, what is the source of the wetland types? Accuracy?
  23. Lines 324 and following. The transfer in-out discussion is not particularly interesting.

In summary, there is considerable amount of useful analysis in this manuscript but it needs more focus and editorial review.  It is too complex and reduction would make it more useful to other scientists.  A discussion of accuracies would be very helpful.